# Increased resting heart rate indicates high-workload hearts with augmented aortic hydraulic power in hypertensive pigs

**Pao-Yen Lin** [1,2] *****, **Bo-Wen Lin**[3], **Tong-Sian Lai**[4], **Yan-Hsiang Yang**[4], **Meei Jyh Jiang**[4] *****

**1** Department of Surgery, College of Medicine, National Cheng Kung University, Tainan, Taiwan, **2** Division of Cardiovascular Surgery, Department of Surgery, National Cheng Kung University Hospital, Tainan, Taiwan, **3** Department of Aircraft Engineering, Air Force Institute of Technology, Kaohsiung, Taiwan, **4** Department of Cell Biology and Anatomy, College of Medicine, National Cheng Kung University, Tainan, Taiwan

***** pylin@mail.ncku.edu.tw (PYL); mjiang@ncku.edu.tw (MJJ)

**Data Availability Statement:** All relevant data are within the manuscript and its Supporting Information files.

## Abstract

Clarifying the inceptive pathophysiology of hypertensive heart disease helps to impede the disease progression. Through coarctation of the infrarenal abdominal aorta (AA), we induced hypertension in minipigs and evaluated physiological reactions and morpho-functional changes of the heart. Moderate aortic coarctation was achieved with approximately 30 mmHg systolic pressure gradient in minipigs. Hypertension was assessed by pressure increment of the carotid artery. Perioperative heart rate (HR) was recorded. We measured aortic flow rate and pressure proximal to coarctation to calculate hydraulic power, an indicator for cardiac workload, and resistance. The hearts harvested at sacrifice were examined for myocardial hypertrophy, fibrosis, and dysfunction. The parameters capable of indicating high-workload heart and their prediction effectiveness were determined by cluster and receiver operating characteristic (ROC) analyses. Prolonged AA coarctation for 8–12 weeks induced hypertension in a portion of minipigs. The cluster of minipigs exhibiting increased aortic hydraulic power displayed hypertension and mean HR elevation without changing arterial resistance. Notably, the blood pressure and HR were measured under full anesthesia, equivalent to resting status. Myocardial hypertrophy was not detected at the tissue, cellular or molecular levels. Expression of biomarkers for cellular stress and heart failure didn't increase except for heat shock protein 40. ROC analysis showed that aortic hydraulic power, resting HR, and mean blood pressure, but not arterial resistance, can serve as the indicators for high-workload hearts. These results suggested that resting HR increase in hypertensive pigs indicates hearts with high workload. Heart failure may develop without appropriate treatment.

## Introduction

Hypertension, with prevalence doubled in the population aged 30–79 years from 1990 to 2019 [1], is a major public health problem. Hypertension may damage vital organs including brain,

**Funding:** The author(s) received no specific funding for this work.

**Competing interests:** The authors have declared that no competing interests exist.

heart, and kidney to cause disorders with high mortality and morbidity. For heart, hypertension induces hypertensive heart disease (HHD) and elicits both structural and functional deterioration of the heart. Untreated HHD ultimately leads to congestive or end-stage heart failure (HF) [2]. According to the guideline of American College of Cardiology and American Heart Association (ACC/AHA), the development of HF is categorized into four stages. Patients of the first 2 stages (A and B) are free of symptom but at risk for developing HF. The risk factors of HF include hypertension, coronary artery disease, diabetes mellitus and so on. Stage A patients demonstrate no structural or functional abnormalities of the left ventricle (LV), i.e. normal or preserved LV systolic function without LV dilatation or hypertrophy (LVH). Stage B patients demonstrate LVH and/or LV dysfunction. Overt structural anomalies can be found among patients of the late stages with symptomatic HF (stage C) or refractory HF (stage D) [3, 4]. HHD is modifiable before impairment of heart structure and function is detected. Therefore, effective management in the early stage can prevent HHD from proceeding to HF.

HHD is a consequence of persistent and uncontrolled hypertension. Classical paradigm suggests that elevated blood pressure (BP) raises the LV afterload, magnifies the myocardial strain, and culminates in LVH which is thought to be the essential morpho-functional pathology of HHD. Pathological LVH with bulky myocyte volume and fibrotic matrix remodeling impairs LV relaxation, increases filling pressure, and induces HF symptoms. When LV is in a state of diastolic dysfunction, ejection fraction (EF) may remain preserved. As the LV myocardia can no longer bear the pressure overload due to sustained hypertension, systolic HF with reduced EF develops. Besides, fatal events such as aberrant conduction arrhythmias and sudden death may occur concomitantly with LVH. However, contrary to classic paradigm, some hypertensive patients bypass the LVH pathway and directly proceed to dilated cardiac failure (with increased LV volume and reduced EF) [5, 6]. In this context, elevated BP instead of LVH is the prerequisite for HHD advancement. Since elevated BP impedes the forward output of stroke volume, LV myocardia have to increase output power to overcome the elevated afterload, thereby resulting in extraordinary cardiac performance. Even without structural alteration, strenuous myocardia are vulnerable to develop HF under such a stressful circumstance. Strenuous hearts usually present some abnormal clinical manifestations. We hence hypothesized that ill physiological signs may appear in the early stage to indicate the commencement of HHD. To verify this hypothesis, a longitudinal observation study using animal model is mandatory.

Several large and small animal models were reported to investigate the pathophysiology of HHD from compensatory LVH to HF. The methods adopted to induce hypertension include aortic banding, aortic stent, aortic cuffing, renal wrap, renal embolization, medication (deoxycorticosterone acetate (DOCA)), and high salt/fat diet. Aortic banding is the most frequently used means for inducing HHD [5]. By handling the extent of aortic constriction, the desired magnitude of LV afterload can be easily achieved. To initiate quick myocardial remodeling, the banding sites are usually located at the ascending aorta and aortic arch close to the LV outflow. However, interventions at proximal aorta may initiate both hypertension and myocardial remodeling in parallel, rendering it difficult to discern the cause-effect relationship between hypertension and LV mal-adaptation. We also used aortic banding to induce hypertension in young adult swine. Through coarctation at the infra-renal abdominal aorta (AA) [7], hypertension was detected 4 to 12 weeks later when blood pressures of brachial artery were measured noninvasively with sphygmomanometer [8]. Therefore, we applied this hypertensive porcine model to investigate the pathophysiological reactions in the early stage of HHD.

This study aims to 1) validate hypertension development in AA coarctation porcine model with standard measurement of intra-arterial pressure; 2) assess the LV workload in response to AA coarctation-induced afterload changes; 3) identify ill physiological signs associated with

strenuous LV in the early stage of HHD; 4) examine myocardial changes at tissue, cellular and molecular levels during the incipient period of hypertension.

## Methods and materials

### Animals

Taiwanese Lanyu (TLY) miniature pigs (7- to 12-month-old adults; both genders) were used in this study. TLY pigs had been qualified as an optimal porcine strain for cardiovascular research [9]. All experimental pigs were provided by Taitung Animal Propagation Station of the Taiwan Livestock Research Institute and received complete quarantine before experiments. Twenty-six miniature pigs were enrolled in this study and randomly assigned into six groups of different time frames: three AA coarctation groups for 4 weeks (4w), 8 weeks (8w) and 12 weeks (12w), and three sham groups (4w, 8w and 12w) as the control. All pigs underwent hemodynamic analyses of right carotid arteries and AAs proximal to regions with coarctation or without (sham). Hemodynamic data for analyses in the carotid arteries included pressures and pulse rates (equivalent to HR) and in the proximal AAs contained volume flow rates, pressures, and HR. Data were obtained at pre- and post-coarctation, and sacrifice. Due to machine calibration and technique problems, pigs receiving hemodynamic analyses of the carotid arteries were nineteen, including 7 in sham group and 12 in the AA coarctation group. The hearts harvested for tissue and protein assays were twenty-four, including 9 in sham group and 15 in the coarctation group. The study conformed to the Guide for the Care and Use of Laboratory Animals published by the National Institutes of Health, and the experimental procedures were approved by the institutional Animal Care and Use Committee of the College of Medicine, National Cheng Kung University.

### Anesthesia and AA coarctation

Anesthesia of TLY pigs was first inducted by intramuscularly injecting a mixture of tranquilizer, analgesics, and anticholinergics containing a mixture of zolazepam and tiletamine (10 mL), xylazine hydrochloride (5 mL), and atropine (1 mL) at the pig's posterior neck or gluteal region. Intravenous route was established at the postauricular vein for infusing parenteral drugs. Endotracheal tube was inserted to assist continuous inhalation of Isoflurane (2% of tidal volume) and maintain general anesthesia during surgery. The sedated pig was taken supine position for conducting laparotomy and trans-peritoneal cavity approach to expose the AA and bilateral common iliac arteries. In aortic banding group, moderate coarctation was executed through an 8 mm-wide Teflon (Polytetrafluoroethylene, PTFE) strip which encircled the AA approximately 2 cm above the aortic bifurcation. The aortic segment wrapped by Teflon strip was then modeled into a funnel-shaped channel with a non-constrictive inlet and a constrictive outlet [7]. Sham group underwent the same procedure without constricting AA by the encircled Teflon strip.

### Quantitative characterization of AA coarctation

A living pig's aorta has the features of continuous pulsation, easy vasospasm and strong resilience against external compression. The nature of porcine aorta makes it hard to quantify the magnitude of aortic constriction by the decrement of luminal area. We hence quantitatively characterized moderate coarctation according to two indicators: systolic aortic pressure gradient (drop) and pulsatility index (PI). A transit time flowmeter system (Medi Stim VeriQ system; MediStim ASA, Oslo, Norway) was applied during operation to simultaneously monitor flow, pressure, and PI changes at the AA regions proximal and distal to coarctation. In the

coarctation group, approximately 30 mm Hg systolic pressure gradient was created between the distal and proximal AA segments. In addition, PI is clinically used to assess the vascular resistance as its reduction is thought to proportionally reflect the severity of aortic coarctation. PI was calculated by dividing the difference between the maximal and minimal flow rate by the mean flow rate [10, 11]. Therefore, we defined moderate coarctation in our porcine model to be PI reduction to one third of the original level at the AA segments proximal and distal to coarctation.

### Blood pressure (BP) measurement

BP was monitored by two systems: 1) noninvasive sphygmomanometer (PAMO II, MEK-ICS, Seongnam-si, Seoul, Korea), which measured BP with pressure cuff (CBP) at one forelimb (brachial artery). BP was recorded every 10 minutes from the induction of anesthesia and through the whole surgery; 2) invasive intra-arterial catheterization (arterial line, ABP), which was established by inserting a 20-gauge catheter into the right common carotid artery. The arterial line connected to a transit time flowmeter system (Medi Stim VeriQ system; MediStim ASA, Oslo, Norway) and continuously recorded the ABP during operation. Systolic, diastolic and mean pressures were all recorded. ABP changes between coarctation and sacrifice were adopted as the basis of hypertension diagnosis, whereas those of CBP were taken as supportive evidence. The reasons for using ABP to identify hypertension are two-fold: 1) ABP is recognized as the gold standard of BP measurement in critical care settings; 2) ABP is more accurate than CBP which is readily affected by the physical and environmental stimuli. These stimuli often produce discrepancies between CBP and ABP [12, 13]. According to the diagnostic criteria of human guideline [14, 15], hypertension in this model was defined as systolic and/or mean BP increase of 20 mmHg and diastolic BP increase of 10 mmHg at sacrifice compared to their counterpart prior to coarctation.

### LV afterload indices: Aortic resistance and aortic hydraulic power

To evaluate the impact of afterload on the LV and the associated physiologic responses, aortic resistance and aortic hydraulic power were analyzed. Pressures and volume flow rates of AA proximal to coarctation were measured by the Medi Stim VeriQ system at pre-coarctation, post-coarctation, and sacrifice. Aortic resistance (mmHg*$min/mL$) was calculated by dividing mean aortic pressure (mmHg) by mean aortic flow rate (mL/min). Aortic hydraulic power is an indicator analogous to cardiac power output (CPO), which is calculated by the product of flow output and systemic arterial pressure dividing 451 [16]. Likewise, we calculated aortic hydraulic power ($mW = mJ/sec$) by the product of mean aortic pressure (mmHg) and mean aortic flow rate (mL/min)) dividing 451.

### Cluster analysis

Aortic hydraulic power exhibited great diversity in pigs with prolonged AA coarctation at sacrifice. Besides, comparing animals within the same coarctation timeframe (e.g. 12-week) are not statistically meaningful because of the insufficient animal number. For further analysis, we applied cluster analysis which is an unsupervised machine learning technique that performs tasks without prior knowledge of the group definition. AA coarctation pigs were clustered into two groups, cluster 1 and cluster 2, based on their hydraulic power values at sacrifice. The analysis was performed by k-means clustering under Python version 3.7 with the module of scikit-learn version 0.24.2.

## Animal sacrifice and tissue collection

Animals were sacrificed with KCl (2 mmol per kilogram body weight) intravenous injection. The hearts were harvested and dry weights measured. A mid-biventricular (2 cm above apex) cross section was made to disclose LV, interventricular septum and right ventricle (RV). The wall thicknesses of LV/RV free walls and interventricular septum at three different sites were measured and averaged. Heart tissues for histological staining were fixed with 10% formalin with one change, followed by dehydration and paraffin embedding. Tissues for western blotting were snap frozen and stored at -80˚C.

## Histological examination of hypertrophy of cardiomyocytes

Paraffin (6 μm thick) were deparaffinized, rehydrated, and stained according to the manufacturer's instructions. The cell cross-sectional area was detected by cardiac myosin heavy chain (MHC) and wheat germ agglutinin (WGA) immunofluorescence staining to assess cardiomyocyte hypertrophy. To calculate the nuclear density of cardiomyocytes, MHC immunofluorescence was conducted, followed by nuclear counterstaining with Hoechst 33342. The nuclear density of cardiomyocytes in a field was calculated by dividing the total MHC-positive area by the number of nuclei. The fluorescence signals were quantified by ImageJ software (v. 1.43u, National Institutes of Health, Bethesda, MD, USA) and Image-Pro Plus software (Media Cybernetics, Inc. USA.).

## Immunoblotting analysis

Equal amounts of protein (40 μg) were resolved on 10% sodium dodecyl sulfate (SDS) gel electrophoresis and transferred to nitrocellulose membranes. Following blocking with Tris Buffered Saline Tween-20 (TBS-T) containing 5% non-fat milk, the membranes were subsequently incubated with anti-myosin light chain 2 (MLC2v) (1:5000; Abcam; ab92721), anti-myosin phosphatase target subunit 1 (MYPT1) (1:2000; Millipore; 30320), anti-MYPT1 Threonine (Thr) 850 (1:1000; Millipore; 36–003), anti-MYPT1 Thr 696 (1:1000; Millipore; 07–251), anti-heat shock protein (HSP) 90 (1:1000; Abcam; ab59459), anti-HSP 70(1:1000; GeneTex; GTX25439), anti-HSP 60 (1:4000; Cell Signaling #12165), anti-HSP 40 (1:500; Abcam; ab223607), anti-osteopontin (1:1000; Abcam; ab231736), anti-alpha-tubulin (1:2500; Novus Biologicals; NB100-069), and with horse radish peroxidase (HRP)-conjugated secondary antibody. The immunoreactive bands were detected with enhanced chemiluminescence kit (ECL; Thermo Scientific, (ECL; Thermo Scientific, 34096 (Femto)), and analyzed with Image J software. The membrane was stripped and reprobed with anti-α-tubulin (1:1000; GeneTex; GTX112141) as the loading control.

## Receiver operating characteristic (ROC) analysis

ROC analysis was performed to evaluate the ability of aortic hydraulic power, resistance, mean heart rate (HR), and mean BP as predictors for detecting sham (control), cluster 1, and cluster 2. Detection performance was represented by indices of the area under the curve (AUC) of ROC, sensitivity, and specificity. The ROC analysis was achieved by machine learning algorithms in Python version 3.7 with the module of scikit-learn version 0.24.2.

## Statistical analysis

Data were expressed as mean ± standard deviation; n indicates the number of pigs. P values were evaluated by one-way analysis of variance (ANOVA) with Tukey post-test, and $P < 0.05$ was considered statistically significant. To examine the correlation between two parameters,

the variance of the correlation coefficient ($R^2$) was obtained through linear or nonlinear regression analysis. Statistical analysis was conducted using GraphPad Prism version 7 (GraphPad Software, San Diego, CA, USA).

## Results

### Prolonged AA coarctation induced hypertension without producing sustained increase in aortic resistance

Hypertension, based on the change of intravascular pressures measured from right carotid artery, was induced by prolonged AA coarctation. Significant increases of systolic, diastolic, and mean carotid arterial pressures were present at 12 weeks post-coarctation (Fig 1A–1C). Noninvasive measurement of brachial arterial pressure with sphygmomanometer also showed significant increases of systolic pressures at 12 weeks post-coarctation (Fig 1D–1F). Aortic resistance, an indicator commonly used to represent LV afterload, was not persistently elevated, however. Significant elevation of aortic resistance occurred immediately following coarctation. At sacrifice, with 4, 8, and 12 weeks of coarctation, aortic resistance of banding groups restored to pre-coarctation levels similar to sham group (Fig 2D). No sustained increase in aortic resistance means that moderate coarctation at distal AA doesn't generate persistent afterload on the LV. On the other hand, hypertension induced by prolonged AA coarctation (8 to 12 weeks in this study) is likely to enhance the afterload of the LV.

### LV workload, reflected by aortic hydraulic power, increased in some pigs undergoing prolonged AA coarctation

CPO has been used as an indicator of cardiac pumping capability [16]. Analogously, using mean aortic pressure and mean aortic flow rate simultaneously measured at AA proximal to coarctation, we calculated aortic hydraulic power. While cardiac output energy decreases

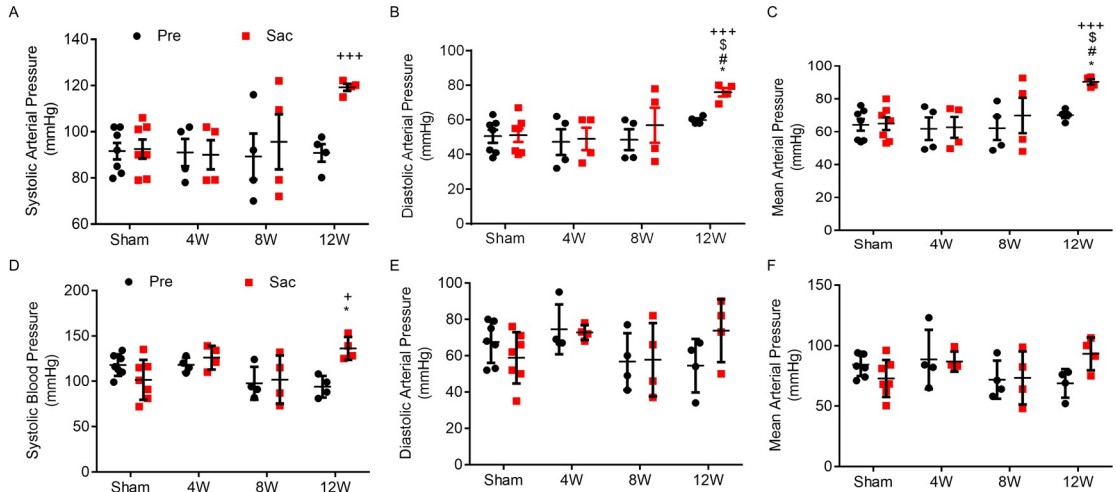

**Fig 1. AA coarctation-induced increases in blood pressure measured by both carotid intra-arterial and brachial cuff methods.** Intra-arterial pressures (ABPs) measured from porcine carotid arteries (A-C): systolic (A), diastolic (B), and mean (C) before coarctation (Pre) and prior to sacrifice (Sac) at 4 weeks (4W), 8 weeks (8W) and 12 weeks (12W). Cuff pressures (CBPs) measured from porcine brachial arteries (D-F): systolic (D), diastolic (E), and mean (F) before coarctation (Pre) and prior to sacrifice (Sac) at 4, 8 and 12 weeks. Sham group, n = 7 (2F, 5M); 4W group, n = 4 (2F, 2M); 8W group, n = 4 (1F, 3M); 12W group, n = 4 (2F, 2M). Data are expressed as mean ± SD, compared to Sham group, * p <0.05, **p<0.01, ***p < 0.001; compared to pre-coarctation, +++ p < 0.001; compared to the 4W group, # p < 0.05; compared to the 8W group, $ p < 0.05. AAC: abdominal aorta coarctation, F: female, M: male.

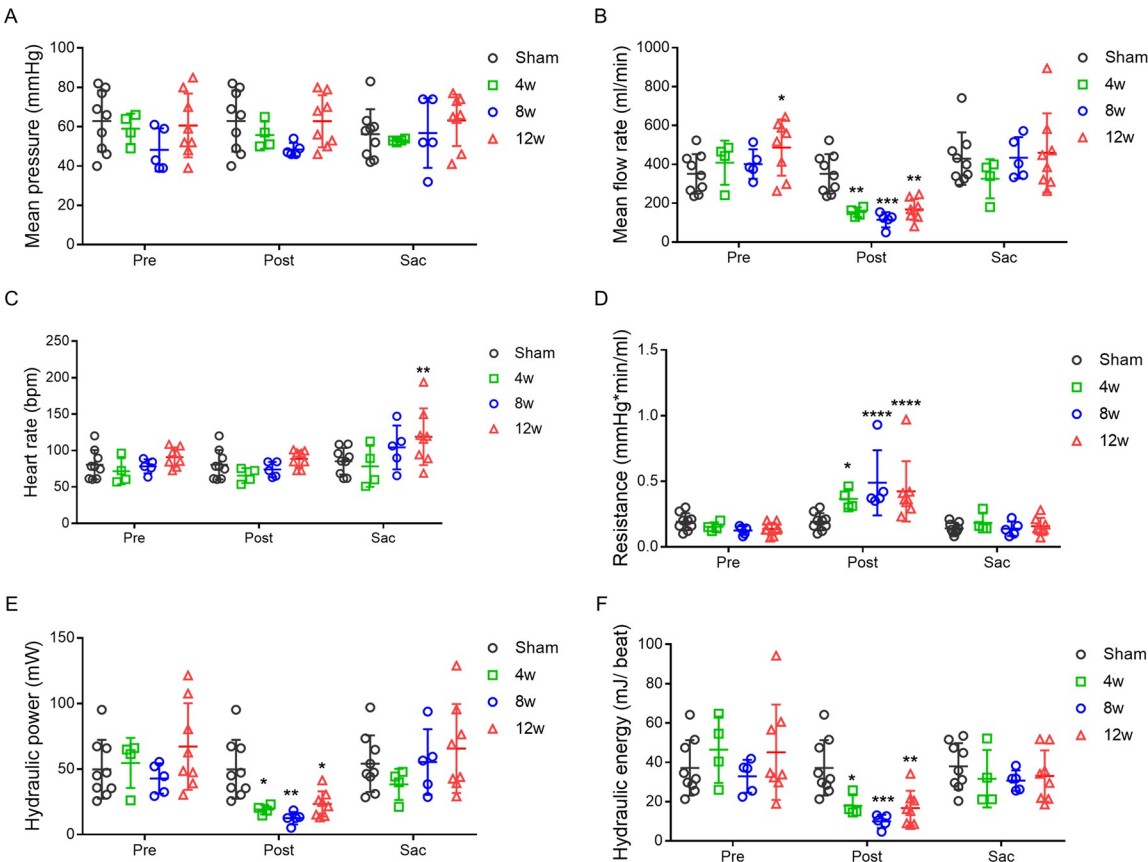

**Fig 2. Hemodynamic analyses of proximal abdominal aortas (AA) following AA coarctation for different time periods.** Mean aortic pressure (A), mean blood flow rate (B), and heart rate (C) were measured in AA proximal to coarctation for pigs undergoing coarctation for 4 weeks (4w), 8 weeks (8w), and 12 weeks (12w) before (Pre) and after (Post) coarctation, and before sacrifice (Sac) using sham group as references. Resistance (D), hydraulic power (E), and hydraulic energy (F) were calculated as described in Methods. Sham group, n = 9 (3F, 6M); 4W group n = 4 (2F, 2M); 8W group n = 5 (1F, 4M); 12W group n = 8 (2F, 6M). Values are expressed as mean ± SD, compared to Sham group, * p < 0.05, ** p < 0.01, *** p < 0.001, **** p< 0.0001. F: female, M: male.

downstream, the aortic hydraulic power at the distal AA substantially reflects the output work of the LV. We therefore used aortic hydraulic power to assess the performance of the heart. During operation, aortic hydraulic power of all banding pigs markedly decreased immediately following AA coarctation. At sacrifice, aortic hydraulic power varied in experimental animals undergoing prolonged AA coarctation. To unveil factors contributing to the observed differences, we applied cluster analysis to all pigs under coarctation for further analysis. Coarctation pigs were hence clustered into two groups, cluster 1 and cluster 2. Pigs of the cluster 1, but not cluster 2, exhibited significant increase of aortic hydraulic power, indicating the augmented output work of the LV. Furthermore, both mean aortic pressure and flow rate increased significantly in the cluster 1 compared to those of sham and cluster 2 (Fig 3).

## Pigs of augmented aortic hydraulic power (cluster 1) exhibited concurrent hypertension and HR increase

Averaged systolic, diastolic and mean carotid arterial pressures of the cluster 1, but not cluster 2 pigs, increased at sacrifice (Fig 4), indicating hypertension in the cluster 1 pigs. Late-developed hypertension added more afterload on LVs of the cluster 1 pigs, thereby compelling

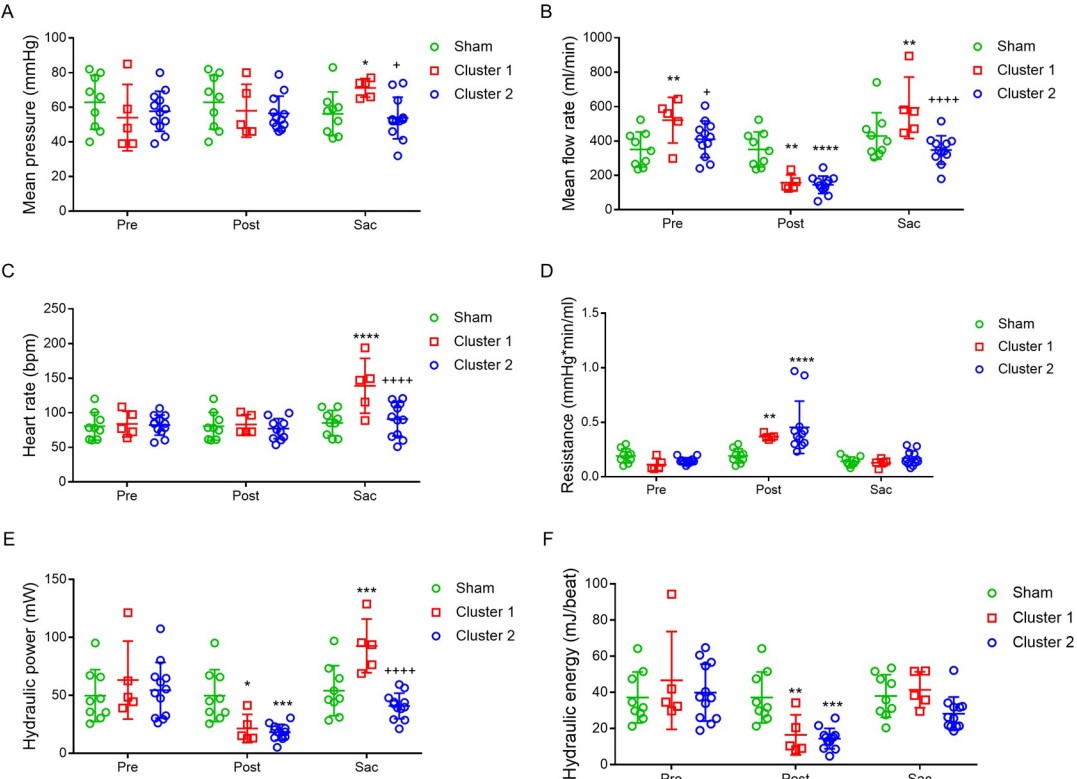

**Fig 3. Hemodynamic analyses of proximal abdominal aortas in pigs exhibiting different aortic hydraulic powers.** Mean aortic pressure (A), mean blood flow rate (B) and heart rate (C) were measured in the abdominal aorta (AA) proximal to coarctation for pigs undergoing AA coarctation before (Pre) and after (Post) coarctation, and before sacrifice (Sac). Resistance (D), hydraulic power (E), and hydraulic energy (F) were calculated as described in Methods. Values of sham group served as references. Pigs undergoing AA coarctation was clustered into two sub-groups based on the presence (Cluster 1) and absence (Cluster 2) of aortic hydraulic power increases at sacrifice. Sham group, n = 9 (3F, 6M); Cluster 1 group, n = 5 (2F, 3M); Cluster 2 group, n = 12, (3F, 9M). Values were expressed as mean ± SD, compared to Sham group, * p < 0.05, ** p < 0.01, *** p < 0.001, **** p < 0.0001; compared to Cluster 1 group, + p < 0.05, ++++ p < 0.0001. F: female, M: male.

hearts to augment the power output for maintaining blood circulation. In contrast, in the absence of sustained increase in resistance and hypertension, the power output of LVs in the cluster 2 pigs was not augmented at sacrifice (Fig 3E). In addition, mean HR at sacrifice was significantly elevated in the cluster 1 group compared with that of the cluster 2 and sham groups (Fig 3C). It is worth noting that mean HR did not differ among the three groups either prior to operation or following AA coarctation. At sacrifice, mean HR remained comparable between the cluster 2 and sham groups. Interestingly, hydraulic energy generated by each heart beat did not differ among the three groups at sacrifice (Fig 3F). This result strongly suggests that hearts of the cluster 1 pigs mainly increase HR to impart adequate hydraulic power for circulation. It is noteworthy that the HR was measured under full anesthesia, and could be esteemed as resting HR. Concurrent elevation of resting HR and aortic hydraulic power in the cluster 1 pigs may represent a physiological reaction to heart stress produced by hypertension.

## Immunohistochemistry and biomarker examination revealed no cellular hypertrophy following AA coarctation

To assess whether LVH occurred, we examined cellular hypertrophy of cardiomyocytes, interstitial fibrosis, and two hypertrophy biomarkers. Cellular hypertrophy was assessed with

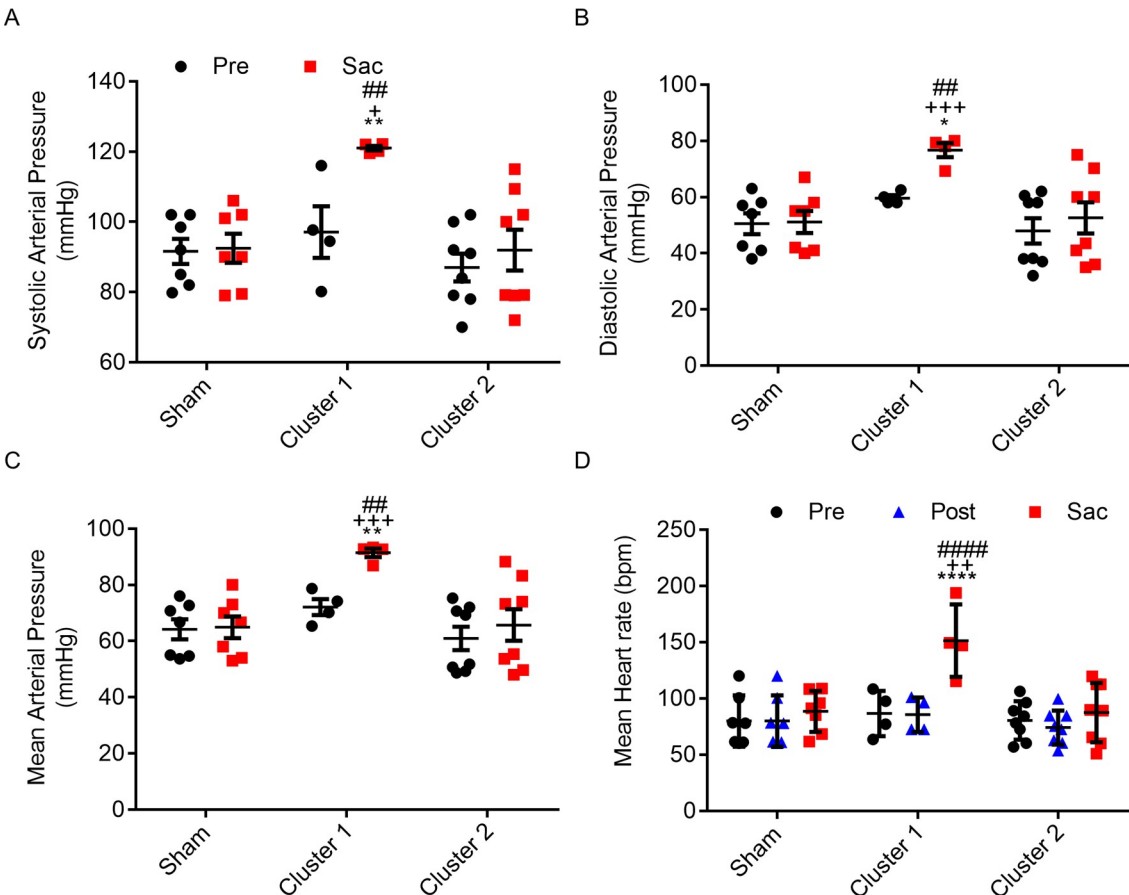

**Fig 4. Cluster 1 pigs exhibited marked increases in both carotid arterial pressures and mean heart rate.** Systolic arterial pressure (A), diastolic arterial pressure (B), mean arterial pressure (C), and mean heart rate (D) were measured in the right carotid artery for pigs undergoing abdominal aorta coarctation before (Pre) and after (Post) coarctation, and before sacrifice (Sac) using sham group as references. Sham group, n = 7 (2F, 5M); Cluster 1 group n = 4 (2F, 2M); Cluster 2 group, n = 8 (3F, 5M). Values were expressed as mean ± SD, compared to Sham group, * $p < 0.05$, ** $p < 0.01$, **** $p < 0.0001$; compared to Pre group, + $p < 0.05$, ++ $p < 0.01$, +++ $p < 0.001$; compared to cluster 2 group, ## $p < 0.01$, #### $p < 0.0001$. F: female, M: male.

cardiomyocyte cross-sectional area and nuclear density using cardiac MHC and WGA immunofluorescence. Within the study period up to 12 weeks following AA coarctation, cellular hypertrophy was not detected in the coarctation groups as compared to sham group (Fig 5 and S1 Fig). We also assessed LVH with activity changes of ventricular myosin light chain 2 (MLC2v) and Rho-associated kinase 1/2 (ROCK 1/2), two well-established markers of cardiac hypertrophy [17, 18]. The activity of MLC2v was assessed with its phosphorylation levels, whereas ROCK activity was accessed with the phosphorylation of myosin phosphatase target subunits 1 and 2 (MYPT1/2). Prolonged AA coarctation up to 12 weeks didn't induce significant activity change in MLC2v or ROCK 1/2 (S2 Fig), consistent with results obtained by cardiomyocyte cross-sectional area and nuclear density analyses.

## Interstitial fibrosis was not detected in the LV following AA coarctation up to 12 weeks

We also examined interstitial fibrosis, another pathological feature of pressure-overload myocardial hypertrophy. Interstitial fibrosis, assessed with Masson's trichrome staining, was not detected in the myocardium of experimental pigs (Fig 5B and 5E).

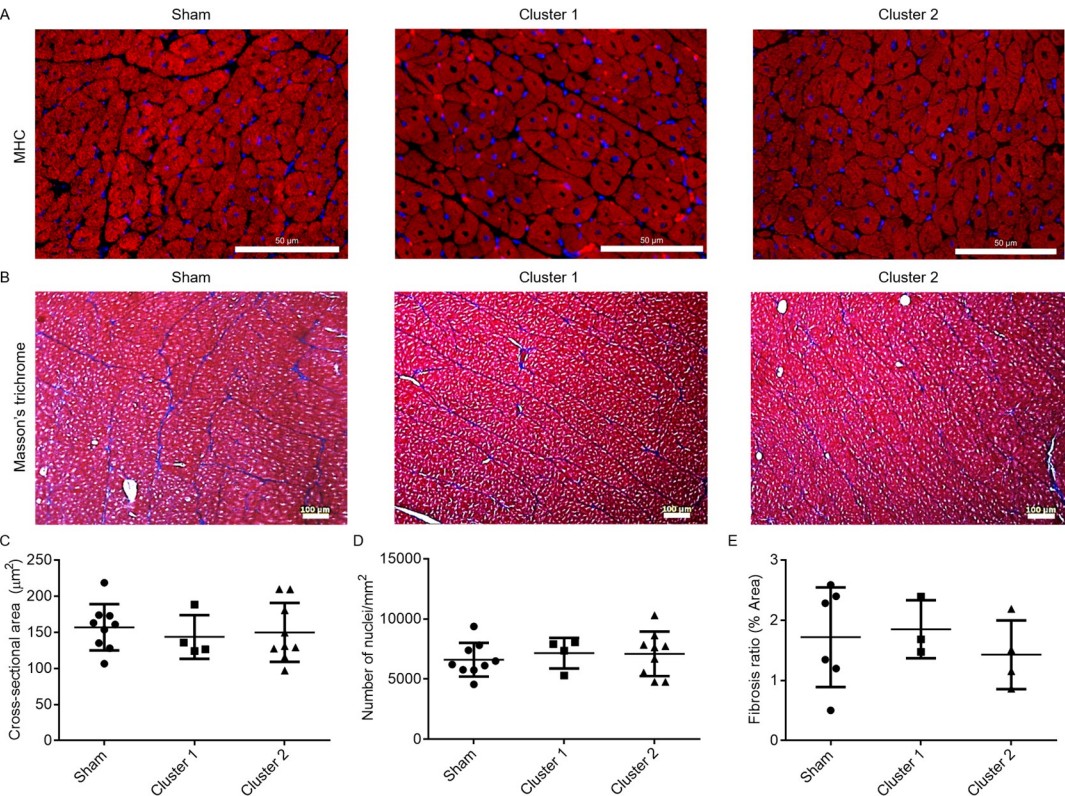

**Fig 5. Cardiomyocyte hypertrophy and myocardial fibrosis were not detected in the left ventricular (LV) wall following AA coarctation.** Immunofluorescence was used to detect cardiomyocytes in tissue sections of the LV of sham (n = 8, 2F and 6M), Cluster 1 (n = 4, 1F and 3M), and Cluster 2 (n = 9, 3F and 6M) groups. Nuclear density of cardiomyocytes was calculated as the number of nuclei per unit area of cardiac myosin heavy chain-positive cells. Myocardial fibrosis in the LV wall was detected with Masson's trichrome staining. Sham group, n = 6 (2F, 4M); Cluster 1 group, n = 3 (1F, 2M); Cluster 2 group, n = 4 (4M). A and B show representative results; C, D, and E summarize quantitative results of cross-sectional area (C), nuclear density (D), and Masson's trichrome staining (E). Values were expressed as mean ± SD. F: female, M: male.

### Expression of heat shock protein 40 (HSP40) was induced in the cluster 1 pigs

To determine the extent of stress response in the heart, we examined the expression of heat shock protein family members including HSP40, HSP60, HSP70, and HSP90 [19]. Marked increase in the expression of HSP40 was detected in the cluster 1 coarctation group compared to sham and cluster 2 groups. No change was detected in HSP60, HSP70, and HSP90. In addition, no change was detected in the expression of osteopontin which was recently shown to be a marker for heart failure (Fig 6) [20].

### Aortic hydraulic power, mean HR, and BP can predict increased LV workload following AA coarctation

As shown in Fig 7A, ROC analysis showed that aortic hydraulic power can distinguish pig hearts with afterload enhancement (cluster 1 and 2) from those without (sham). Marked augmentation of aortic hydraulic power reflects strenuous work executed by myocardium following AA coarctation. Clearly, aortic hydraulic power can also distinguish strenuous hearts (cluster 1) from non-strenuous hearts (cluster 2) (Fig 7A). In contrast, resistance is not effective in predicting any of the three groups (Fig 7B). Because HR and MAP were markedly

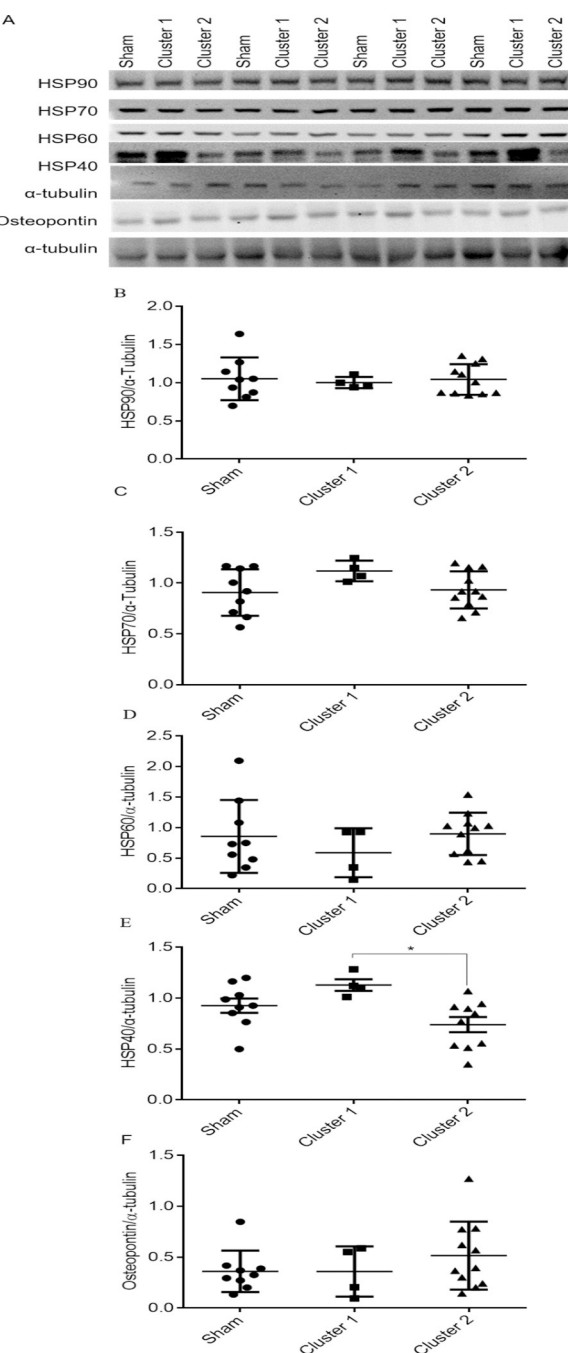

**Fig 6. The expression of heat shock protein 40 was upregulated in the left ventricle (LV) of the Cluster 1 group.**
The expression levels of heat shock protein 90 (HSP90), HSP70, HSP60, HSP40, and osteopontin in the LV wall were detected with immunoblotting using α-tubulin as the loading control. A shows representative immunoblots and B-F summarize quantitative results of HSP90 (B), HSP70 (C), HSP60 (D), HSP40 (E), and osteopontin (F). Sham group, n = 9 (3F, 6M); Cluster 1 group, n = 4 (2F, 2M); Cluster 2 group, n = 11 (3F, 8M). Values are expressed as mean ± SD, * p < 0.05 Cluster 1 vs. Cluster 2. F: female, M: male.

increased in the cluster 1, we also analyzed their applicability as indicators to distinguish the cluster 1 pigs from others. Both mean HR and mean BP demonstrate excellent effectiveness in identifying the cluster 1 group with high aortic hydraulic power (Fig 7C and 7D). Taken

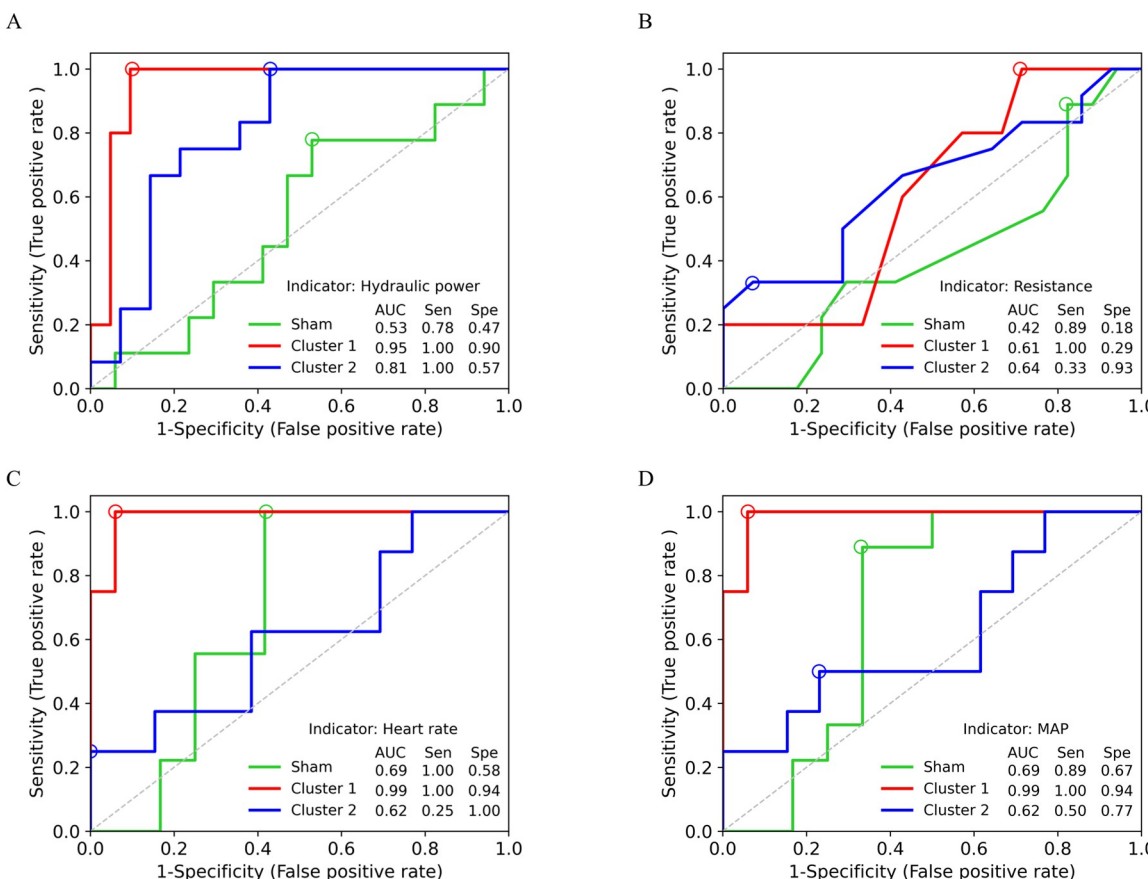

**Fig 7.** Receiver operating characteristic (ROC) curves for prediction of sham, Cluster 1, and Cluster 2 groups with indicators hydraulic power (A), resistance (B), mean heart rate (C), and mean arterial pressure (D). The optimal values of sensitivity and specificity for each ROC curve, as indicated with circles, were selected based on the Youden index. AUC: area under ROC curve; Sen: sensitivity; Spe: specificity.

together, increase in aortic hydraulic power, resting HR, and mean BP can serve as the indicator to survey the heart condition.

## Discussion

Prolonged moderate coarctation at the infrarenal AA induced hypertension with increased resting HR in swine. The concurrent increase in BP and resting HR contributes to higher aortic hydraulic power that indicates elevated workload for the LV. In this study, BPs and HRs were measured under full anesthesia with stable hemodynamics. Therefore, the perioperative physiological and environmental stresses are likely to cause minimal interferences in BP and HR measurements. Our results showed that systolic, diastolic, and mean BPs measured by ABP all increased significantly in pigs at 12 weeks post-coarctation. ABPs of some pigs also markedly elevated at 8 weeks post-coarctation (Fig 1). Changes of ABP, the gold standard of BP measurement [12, 13], between coarctation and sacrifice were adopted for the diagnosis of hypertension. Home BP, measured under resting state free of environmental and emotional stress, is endorsed as the fundamental of human hypertension diagnosis. In human, data of home BP monitoring (HBPM) are correlated with hypertension-mediated organ damage and cardiovascular events. Based on HBPM, the commonly used definition and grading of hypertension in adults are as follows: optimal BP is systolic < 120 mmHg and diastolic < 80 mmHg;

normal is systolic 120–129 mmHg and diastolic < 80 mmHg; definite hypertension is systolic ≥ 140 mmHg or diastolic ≥ 90 mmHg [14, 15]. A difference of approximately 20 mmHg in systolic pressure and 10 mmHg in diastolic pressure is found between normal BP and definite hypertension. Therefore, we defined hypertension in our anesthetized pigs as the increments of systolic pressure > 20 mmHg and diastolic pressure > 10 mmHg. Our results showed that significant increase of systolic pressure > 30 mmHg, diastolic pressure > 10 mmHg, and mean pressure > 20 mmHg were detected in ABPs of all pigs at 12w post-coarctation and some pigs at 8w post-coarctation compared to that of pre-coarctation and sham group (Fig 1A–1C). Accordingly, definite hypertension developed in our AA coarctation porcine model. Aortic hydraulic power, representing LV workload, also increased as AA coarctation prolonged (Fig 2E). The temporal association between hypertension and increased aortic hydraulic power suggests that late-developed hypertension amplifies LV afterload. Moreover, HR increased with a trend similar to aortic hydraulic power (Fig 2C), suggesting that HR increase may indicate the rising LV workload.

Suprarenal aortic banding is the mainstay of hypertensive animal models [5, 21–23]. These models promptly actuate BP elevation resembling renovascular hypertension which mainly involves hypoperfusion to the kidneys and hyperactivation of the renin-angiotensin-aldosterone system (RAAS) [24, 25]. On the other hand, chronic or stepwise narrowing of the aortic arch, ascending or descending thoracic aorta dampens sensitivity of arterial baroreflex and triggers delayed hypertensive response [26]. In our model, pigs received AA coarctation approximately 3–4 cm below the orifices of renal arteries. Aortic constriction at this site is unlikely to compromise renal perfusion. Subsequent measurements showed that hypertension didn't happen within a short period after aortic banding. No BP elevation was observed in ABP measurements of all pigs at 4w post-coarctation and the majority of pigs at 8w post-coarctation (Fig 1A). These results indicated that hypertension in this model was not initiated by RAAS activation. Moreover, baroreceptors at carotid sinus and aortic arch can quickly sense pressure changes of the thoracic aorta, and further activate arterial baroreflex to regulate BP, particularly mean arterial pressure (MAP) which is crucial for tissue perfusion. In addition, HR is regulated to maintain sufficient cardiac output (CO) [27]. Carotid MAP and HR exhibited no difference between pre- and post-coarctation, but both displayed significant increases at 12w post-coarctation (Figs 1B and 2C). Late-developed hypertension with high HR implies that prolonged AA coarctation might blunt the function of arterial baroreflex.

Hypertension causes pressure overload on the LV, magnifies myocardial strain, and ultimately leads to HF. Previous swine models of pressure overload were usually fulfilled by banding ascending aorta to produce a pronounced systolic pressure gradient of 50–80 mmHg, by which magnified LV afterload would induce HF within two to six months [28]. Moderate coarctation at the infrarenal AA with an approximate 30-mmHg systolic pressure gradient didn't evoke HF signs at 12 weeks post-coarctation. Instead, hypertension was detected via carotid ABP in a portion of pigs undergoing coarctation, resulting in pressure overload. These results showed that moderate AA coarctation elicited hypertension preceding to HF. Aortic resistance, a commonly used index of LV afterload, was not persistently elevated following AA coarctation. Under sustained banding, aortic resistance at AA proximal to coarctation increased at immediate post-coarctation and then returned to baseline at 4w, 8w, and 12w post-coarctation (Fig 2D). The volume flow rate measured at the same region, mimicking CO, fell temporarily following coarctation and recovered during the ensuing period (Fig 2B). No change of mean aortic pressure was observed in the coarctation group at pre-coarctation, post-coarctation, and different sacrifice timepoints (Fig 2A). Of note, aortic resistance was calculated by dividing mean aortic pressure by mean aortic flow rate. Return of aortic resistance under unaltered aortic pressure suggests that CO is increased to compensate for the afterload

augmentation. Based on Anrep effect, myocardial contractility is stimulated by increased afterload to adapt to output impedance [29, 30]. It is appropriate to assess the LV pumping work in response to the magnified LV afterload while the heart systolic function is preserved.

Heart systole, through the generation of both flow (CO) and pressure, imparts hydraulic energy into arterial system to maintain blood circulation. CPO, calculated by the product of MAP and CO dividing 451, is an indicator representing the ventricular pumping capability [16]. Aortic hydraulic power calculated by the product of mean pressure and flow rate measured at AA is analogous to CPO. Therefore, we used aortic hydraulic power to assess the LV work in response to afterload magnification. Mean aortic hydraulic power markedly declined at immediate post-coarctation but rose steadily as AA coarctation prolonged (Fig 2E), suggesting that LVs increased pumping work to overcome output impedance and maintain adequate CO. To clarify the correlation of hypertension with myocardial workload, we clustered the coarctation group based on aortic hydraulic power at sacrifice. Pigs with AA coarctation were distinguished into two clusters with cluster 1 exhibiting augmented aortic hydraulic power and cluster 2 without (Fig 3E). Mean volume flow rate, reflecting CO, at AA proximal to coarctation rose significantly in the cluster 1 group (Fig 3B). These results suggested that LVs of the cluster 1 pigs were under high workload. In this context, carotid systolic pressure, diastolic pressure, and MAPs of the cluster 1, but not cluster 2 pigs, increased at sacrifice (Fig 4A–4C), indicating hypertension in the cluster 1 pigs. Apparently, late-developed hypertension increased afterload on LVs of the cluster 1 pigs, thereby compelling hearts to augment the power output for maintaining blood circulation. Contrary to aortic hydraulic power, aortic resistance failed to distinguish the hearts with high workload from others. The values of aortic resistance among sham, cluster 1, and cluster 2 groups were all close to the baseline at sacrifice (Fig 3D). These results point out that aortic resistance is not a good indicator for assessing LV afterload.

Resting HR is an accepted biomarker of sympathetic cardiovascular drive which plays a major role in hypertension [31]. Under anesthesia, mean HR is equivalent to resting HR. In this study, mean HR at sacrifice increased significantly in the cluster 1 group (Figs 3C and 4D), suggesting that HR increases may contribute to rising LV workload. Aortic hydraulic energy displays output energy generated with each heartbeat and is analogous to stroke volume. Mean aortic hydraulic energy values were not different among three groups at sacrifice, all close to those at pre-coarctation (Fig 3F). This result was consistent with the results of four-chamber image echocardiography conducted at sacrifice via sub-xiphoid approach to assess stroke volume of LV in some pigs. While the results were not validated due to small sample size and lack of pre-coarctation data, we detected no difference in LV stroke volume at sacrifice among sham, cluster 1, and cluster 2 groups. Based on CO = HR x stroke volume, we inferred that high CO occurring in the cluster 1 group is attributed to rapid HR. In this context, European Society of Hypertension (ESH) guidelines stated that hypertensive patients with rapid resting HR (>80 beat per minute) are predicted to possess high cardiovascular risk [14, 31, 32]. The detection of increased HR in the cluster 1 pigs may strengthen the link between rapid resting HR and HHD inception. Rapid resting HR reflects high workload of the LV. Strenuous heart, even without structural alteration or impaired contractility, is vulnerable to develop HF under sustained hypertensive stress.

Advanced HHD are characterized with anomalies at tissue, cellular, and molecular levels. To evaluate whether structural anomalies of LV developed, we examined several hypertrophy features, including cardiomyocyte hypertrophy, myocardial fibrosis, and biomarkers of myocardial hypercontractility [33]. Our results consistently showed that LVH did not occur in the coarctation group. Cardiomyocyte cross-sectional area and nuclear density measurements both indicated no cellular hypertrophy in pigs undergoing coarctation compared to sham

group. In addition, interstitial fibrosis was not detected in either cluster 1 or cluster 2 groups (Fig 5 and S1 Fig). We also examined whether coarctation produced changes in activities of MLC2v and ROCK 1/2, both being recognized biomarkers for stress-induced LVH [17, 18]. No change was detected with either biomarker, suggesting that myosin-mediated hypercontractility did not occur in the cluster 1 pigs exhibiting high CO. In addition, the heart weight-to-body weight ratio and thickness ratio between free LV wall/interventricular septum wall and the right ventricle wall indicated no change among groups (Lai TH, Yang YH, Lin PY, and Jiang MJ, unpublished results). Taken together, our results suggested that distal AA coarctation up to 12 weeks did not induce LVH. Furthermore, we detected no increase in the expression of osteopontin which was proposed to downregulate cardiac bioenergetics in HF patients with hypertension and diastolic dysfunction [20]. These results indicate that pigs undergoing AA coarctation up to 12 weeks do not develop diastolic HF. Taken together, the cluster 1 pigs in this study can be classified as stage A HHD without definite structural anomalies.

Hypertension enhances LV afterload and magnifies myocardial strain. Upregulation of heat shock protein synthesis upon environmental stress assists cardiomyocytes to maintain cellular protein homeostasis and to ensure cell survival. Previous studies indicated that HSP40, HSP60, HSP70, and HSP90 play roles in cardiac protection against stresses [19, 34]. Interestingly, the expression of HSP40 markedly increased in the cluster 1 group whereas that of other HSPs did not (Fig 6). Given that the cluster 1 pigs displayed hypertension with augmented aortic hydraulic power, these results imply that HSP40 may be upregulated in strenuous myocardia to counteract further damages caused by hypertensive stress. Nonetheless, persistent hypertensive stress may propel the progression of pathological LV remodeling.

ROC analysis is useful in clinical medicine to determine whether a test has the capacity to discriminate between diseased and non-diseased states [35]. We used ROC analysis to evaluate the ability of aortic hydraulic power, resistance, HR, and MAP as predictors for detecting sham (control), cluster 1, and cluster 2. ROC curve showed that aortic hydraulic power is capable of distinguishing AA coarctation pigs (cluster 1 and 2) from sham group. Moreover, aortic hydraulic power is effective in distinguishing strenuous hearts (cluster 1) from non-strenuous hearts (cluster 2) (Fig 7A). In contrast, resistance is not effective in predicting any of the three groups (Fig 7B). Aortic hydraulic power is calculated by aortic pressure and flow rate, both are impossible to self-monitor in general circumstances. Therefore, we analyzed two systemic physiological parameters, MAP and HR, to assess their abilities for detecting strenuous hearts. ROC showed that both HR and MAP exhibit excellent efficacy in identifying the cluster 1 group with high aortic hydraulic power (Fig 7C and 7D). As a result, HR and MAP can be used as independent indicators to monitor HHD progression.

This study is an event-driven prospective study that used AA coarctation to induce hypertension. Definite hypertension measured by ABP did occur in pigs with prolonged AA coarctation around 12 weeks (Fig 1). Notably, the majority of pigs exhibiting increased aortic hydraulic power, i.e. cluster 1, also underwent AA coarctation for 12 weeks. The temporal association between hypertension and elevated aortic hydraulic power implies that AA coarctation, if applied long enough, results in strenuous heart under hypertensive stress. Strenuous hearts are thought to further magnify myocardial strain, culminate in LVH, and lead to LV dysfunction. Our experimental period was set for 12 weeks, which may not provide enough time for enhanced afterload exerted by newly-developed hypertension to elicit structural deterioration of the LV. Longer study period is needed to clarify the pathological progression of HHD. Significant HR increase in the cluster 1 pigs is the highlight of this study, which suggests that elevated resting HR indicates strenuous heart in the early stage of HHD. Hypertension without appropriate treatment may lead to HF.

## Conclusion

Prolonged moderate coarctation of porcine infrarenal AA, around 8–12 weeks, induced hypertension and HR increase, thereby increasing workload for the LV. Aortic hydraulic power, analogous to cardiac power output, is an indicator for LV workload. Accordingly, pigs undergoing AA coarctation were regrouped into clusters with and without increase in aortic hydraulic power. Albeit no sign of LV hypertrophy or HF detected, the cluster 1 pigs with increased aortic hydraulic power displayed hypertension and high resting HR compared to the cluster 2 and sham groups. ROC analysis indicates that in addition to aortic hydraulic power, HR and MAP are both highly effective in distinguishing the cluster 1 pigs from others. These results suggest that concurrence of high resting HR, high hydraulic power, and hypertension indicates high-workload LV myocardia which may proceed to HF.

## Supporting information

**S1 Fig. Cross-sectional area of cardiomyocytes in the left ventricle (LV) does not change following AA coarctation.** The cross-sectional area of cardiomyocytes was assessed with fluorescent images of wheat germ agglutinin (WGA)- and Hoechst 33342-stained cells. Sham group, n = 8 (2F, 6M); Cluster 1 group, n = 2 (1F, 1M); Cluster 2 group, n = 9 (3F, 6M). Data are expressed as mean ± SD. F: female, M: male.
(TIF)

**S2 Fig. Activation levels of cardiac myosin light chain 2 (MLC2v) and Rho-associated kinase 1/2 (ROCK 1/2) in the LV wall following AA coarctation.** Activation levels of two cardiac hypertrophy biomarkers, MLC2v and ROCK 1/2, were examined in the left ventricle (LV) of sham, Cluster 1, and Cluster 2 groups. MLC2v activation was assessed with Phos-tag SDS-PAGE with subsequent immunoblotting to detect the phosphorylated and non-phosphorylated MLC2v (A). Sham group, n = 9 (3F, 6M); Cluster 1 group, n = 5 (2F, 3M); Cluster 2 group, n = 10 (3F, 7M). ROCK 1/2 activation was assessed with the phosphorylation levels (T696 and T850) of myosin phosphatase target subunit 1 (MYPT1, B). Sham group, n = 9 (3F, 6M); Cluster 1 group, n = 4 (2F, 2M) for T696, n = 5 (2F, 3M) for T850; Cluster 2 group, n = 11 (3F, 8M). Data are expressed as mean ± SD. F: female, M: male.
(TIF)

**S1 Raw data.**
(XLSX)

**S1 Raw images.**
(PDF)

## Acknowledgments

We thank the technical services provided by the "Bioimaging Core Facility of the National Core Facility for Biopharmaceuticals, National Science and Technology Council, Taiwan, and the support of the Core Laboratory, Clinical Medicine Research Center, National Cheng Kung University Hospital.

## Author Contributions

**Conceptualization:** Pao-Yen Lin, Meei Jyh Jiang.

**Data curation:** Pao-Yen Lin, Tong-Sian Lai, Yan-Hsiang Yang, Meei Jyh Jiang.

**Formal analysis:** Pao-Yen Lin, Bo-Wen Lin, Meei Jyh Jiang.

**Funding acquisition:** Pao-Yen Lin, Meei Jyh Jiang.

**Investigation:** Pao-Yen Lin, Meei Jyh Jiang.

**Methodology:** Pao-Yen Lin, Bo-Wen Lin, Tong-Sian Lai, Yan-Hsiang Yang, Meei Jyh Jiang.

**Project administration:** Pao-Yen Lin, Yan-Hsiang Yang, Meei Jyh Jiang.

**Resources:** Pao-Yen Lin, Meei Jyh Jiang.

**Software:** Bo-Wen Lin.

**Supervision:** Pao-Yen Lin, Meei Jyh Jiang.

**Validation:** Pao-Yen Lin, Bo-Wen Lin, Yan-Hsiang Yang, Meei Jyh Jiang.

**Visualization:** Pao-Yen Lin, Bo-Wen Lin, Yan-Hsiang Yang, Meei Jyh Jiang.

**Writing – original draft:** Pao-Yen Lin, Bo-Wen Lin, Tong-Sian Lai, Yan-Hsiang Yang, Meei Jyh Jiang.

**Writing – review & editing:** Pao-Yen Lin, Bo-Wen Lin, Meei Jyh Jiang.

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
