## [Decision Letter · Decision Letter 0]

9 Jul 2024

PONE-D-24-23306Increased resting heart rate indicates strenuous hearts with augmented aortic hydraulic power in hypertensive pigsPLOS ONE

Dear Dr. Lin,

Thank you for submitting your manuscript to PLOS ONE. After careful consideration, we feel that it has merit but does not fully meet PLOS ONE’s publication criteria as it currently stands. Therefore, we invite you to submit a revised version of the manuscript that addresses the points raised during the review process.

We look forward to receiving your revised manuscript.

Kind regards,

Masaki Mogi

Academic Editor

PLOS ONE

Journal Requirements:

Additional Editor Comments:

Major revisions are necessary in the present form. See the Reviwers’ comments.

Reviewers' comments:

Reviewer's Responses to Questions

**Comments to the Author**

1. Is the manuscript technically sound, and do the data support the conclusions?

Reviewer #1: Partly

Reviewer #2: Partly

2. Has the statistical analysis been performed appropriately and rigorously? 

Reviewer #1: Yes

Reviewer #2: Yes

3. Have the authors made all data underlying the findings in their manuscript fully available?

Reviewer #1: No

Reviewer #2: Yes

4. Is the manuscript presented in an intelligible fashion and written in standard English?

Reviewer #1: Yes

Reviewer #2: Yes

5. Review Comments to the Author

Reviewer #1: Although the topic is interesting, the study is hard to follow in its current form. I have the following comments for the authors to consider.

The title suggests (incorrectly) that increased heart rate is a cause of augmented aortic hydraulic power. Please rewrite accordingly.

The introduction is broad and hard to follow. Overall, the authors should better highlight the current gaps in the literature and the rationale for the present study.

Notably, the first aim of the study was to "clarify whether hypertension develops in AA coarctation porcine model". However, the authors stated in lines 113-115 that they already showed the effectiveness of their model in inducing hypertension (PMID: 29615895). In fact, Figure 1 of both papers is very similar (if not identical). Did you use the same animals? Please clarify.

Lines 116-120: What does Alzheimer's disease have to do with this study?

The time frames for the cluster analysis are not clear. Was it at 8 or 12 weeks? The authors stated that coarctation induced hypertension in all animals at week 12. However, in the cluster analysis, only animals in cluster 1 displayed increases in blood pressure. This is confusing.

Lines 288-290: Didn’t you measure aortic resistance at weeks 8 and 12?

Figures: Please provide the sample size and the number of males/females per group in the figure legends.

It is recommended to show the data as mean � SD in all figures.

It is recommended to include a limitation and perspective/significance section in the discussion.

Reviewer #2: Lin and colleagues determined the pathophysiological changes of the heart in response to hypertension induced by infrarenal abdominal aorta coarctation in miniature pigs. The authors found that 12 weeks of moderate aortic coarctation increased systolic arterial pressure, mean arterial pressure, and heart rate. By clustering the animals into two groups based on the hydraulic power value in the aorta at sacrifice, the author reported that the group with increased aortic hydraulic power demonstrated hypertension and high heart rate compared to the other group and sham controls. Overall, this manuscript could be novel in understanding AA coarctation-induced hypertension as a potential model for studying the pathophysiology of hypertensive heart disease.

Points of concern are outlined below:

Major:

1. The application of coarctation of the infrarenal abdominal aorta (AA) to induce hypertension and subsequent left ventricle (LV) hypertrophy in the current manuscript is questionable. The increase in arterial pressure and heart rate was only observed 12 weeks post-surgery (Fig 1 and Fig 2C), while all parameters assessing LV hypertrophy and myocardial fibrosis were not seen (Fig 5). In addition, the mean pressure in the aorta shows no difference pre- and post-surgery and at sacrifice (Fig 2A). These observations suggest the AA coarctation was either not sufficient or was not applied long enough to induce LV hypertrophy.

2. Why are the diastolic BP and MBP in 8wk and 12wk groups before surgery (Fig 1 E&F) lower than other groups as measured by brachial cuff methods?

3. The criteria used to separate clusters 1 and 2 are inappropriate. While cluster 2 consists of animals from all three treatment groups, 4 of 5 animals in cluster 1 are from the 12w group. Considering that the onset of hypertension was observed around 12 weeks post-surgery (Fig 1), the author should only classify and compare the cluster of animals from the same treatment group.

4. The mean pressure, hydraulic power, and hydraulic energy before surgery and at sacrifice were not different among all groups (Fig 2 A, E&F). Given the critique on the cluster classification above, the application of aortic hydraulic power and MAP&HR to predict HHD progression should be verified in a separate group of animals.

5. Why is the n in Figure 4 less than the one from Figure 3? Were carotid arterial pressure and heart rate only measured in selected animals in Figure 4? Or were the data of some animals excluded? Please provide justification if some data were excluded.

6. Is the n of cluster 1 in Figure 5C equal to 2? If so, the authors should increase the number of samples in cluster 1 for LV hypertrophy assessment.

Minor:

1. Please add n of all groups in all figures.

6. PLOS authors have the option to publish the peer review history of their article (what does this mean?). If published, this will include your full peer review and any attached files.

Reviewer #1: No

Reviewer #2: No

---

## [Author Response · Author response to Decision Letter 0]

12 Nov 2024

To academic editor and reviewers:

Your recommendation and comments had been well taken. We've made the revision following these suggestion and re-submitted the new version of manuscript. Hope the revised manuscript meets your requirement and can be accepted for publication.

Best Regards

Pao-Yen Lin MD, PhD

---

## [Decision Letter · Decision Letter 1]

27 Nov 2024

PONE-D-24-23306R1Increased resting heart rate indicates high-workload hearts with augmented aortic hydraulic power in hypertensive pigsPLOS ONE

Dear Dr. Lin,

Thank you for submitting your manuscript to PLOS ONE. After careful consideration, we feel that it has merit but does not fully meet PLOS ONE’s publication criteria as it currently stands. Therefore, we invite you to submit a revised version of the manuscript that addresses the points raised during the review process.

There are still minor revisions in the present form. 

We look forward to receiving your revised manuscript.

Kind regards,

Masaki Mogi

Academic Editor

PLOS ONE

Journal Requirements:

Reviewers' comments:

Reviewer's Responses to Questions

**Comments to the Author**

1. If the authors have adequately addressed your comments raised in a previous round of review and you feel that this manuscript is now acceptable for publication, you may indicate that here to bypass the “Comments to the Author” section, enter your conflict of interest statement in the “Confidential to Editor” section, and submit your "Accept" recommendation.

Reviewer #1: All comments have been addressed

Reviewer #2: All comments have been addressed

2. Is the manuscript technically sound, and do the data support the conclusions?

Reviewer #1: Yes

Reviewer #2: Yes

3. Has the statistical analysis been performed appropriately and rigorously? 

Reviewer #1: Yes

Reviewer #2: Yes

4. Have the authors made all data underlying the findings in their manuscript fully available?

Reviewer #1: Yes

Reviewer #2: Yes

5. Is the manuscript presented in an intelligible fashion and written in standard English?

Reviewer #1: Yes

Reviewer #2: Yes

6. Review Comments to the Author

Reviewer #1: The authors have adequately addressed my comments, and the manuscript has indeed improved. I have no further concerns.

Reviewer #2: Lin and colleagues determined the pathophysiological changes of the heart in response to hypertension induced by infrarenal abdominal aorta coarctation in miniature pigs. The authors found that 12 weeks of moderate aortic coarctation increased systolic arterial pressure, mean arterial pressure, and heart rate. By clustering the animals into two groups based on the hydraulic power value in the aorta at sacrifice, the author reported that the group with increased aortic hydraulic power demonstrated hypertension and high heart rate compared to the other group and sham controls. Overall, this manuscript could be novel in understanding AA coarctation-induced hypertension as a potential model for studying the pathophysiology of hypertensive heart disease.

Comment to Response #3 to Reviewer 2: Please add this explanation of how cluster analysis is performed to line 289 of the revised manuscript. This information would solve the confusion of how these two clusters are classified.

7. PLOS authors have the option to publish the peer review history of their article (what does this mean?). If published, this will include your full peer review and any attached files.

Reviewer #1: No

Reviewer #2: No

---

## [Author Response · Author response to Decision Letter 1]

10 Dec 2024

Response to reviewer 2: Thank you for your suggestion. To clearly convey the idea on cluster analysis, we changed the manuscript in two places, lines 177-181 in Methods section and lines 293-295 in Results section.

---

## [Decision Letter · Decision Letter 2]

15 Dec 2024

Increased resting heart rate indicates high-workload hearts with augmented aortic hydraulic power in hypertensive pigs

PONE-D-24-23306R2

Dear Dr. Lin,

We’re pleased to inform you that your manuscript has been judged scientifically suitable for publication and will be formally accepted for publication once it meets all outstanding technical requirements.

Kind regards,

Masaki Mogi

Academic Editor

PLOS ONE

Additional Editor Comments (optional):

Reviewers' comments:

Reviewer's Responses to Questions

**Comments to the Author**

1. If the authors have adequately addressed your comments raised in a previous round of review and you feel that this manuscript is now acceptable for publication, you may indicate that here to bypass the “Comments to the Author” section, enter your conflict of interest statement in the “Confidential to Editor” section, and submit your "Accept" recommendation.

Reviewer #2: All comments have been addressed

2. Is the manuscript technically sound, and do the data support the conclusions?

Reviewer #2: Yes

3. Has the statistical analysis been performed appropriately and rigorously? 

Reviewer #2: Yes

4. Have the authors made all data underlying the findings in their manuscript fully available?

Reviewer #2: Yes

5. Is the manuscript presented in an intelligible fashion and written in standard English?

Reviewer #2: Yes

6. Review Comments to the Author

Reviewer #2: Lin and colleagues determined the pathophysiological changes of the heart in response to hypertension induced by infrarenal abdominal aorta coarctation in miniature pigs. The authors found that 12 weeks of moderate aortic coarctation increased systolic arterial pressure, mean arterial pressure, and heart rate. By clustering the animals into two groups based on the hydraulic power value in the aorta at sacrifice, the author reported that the group with increased aortic hydraulic power demonstrated hypertension and high heart rate compared to the other group and sham controls. Overall, this manuscript could be novel in understanding AA coarctation-induced hypertension as a potential model for studying the pathophysiology of hypertensive heart disease. The authors have addressed all my concerns. I have no further comments.

7. PLOS authors have the option to publish the peer review history of their article (what does this mean?). If published, this will include your full peer review and any attached files.

Reviewer #2: No

---

## [Editor Report · Acceptance letter]

3 Jan 2025

PONE-D-24-23306R2 

PLOS ONE

Dear Dr. Lin, 

I'm pleased to inform you that your manuscript has been deemed suitable for publication in PLOS ONE. Congratulations! Your manuscript is now being handed over to our production team.

Kind regards, 

on behalf of

Dr. Masaki Mogi 

Academic Editor

PLOS ONE